# Anti-CK7/CK20 Immunohistochemistry Did Not Associate with the Metastatic Site in TTF-1-Negative Lung Cancer

**DOI:** 10.3390/diagnostics12071589

**Published:** 2022-06-29

**Authors:** Alice Court, David Laville, Sami Dagher, Vincent Grosjean, Pierre Dal-Col, Violaine Yvorel, François Casteillo, Sophie Bayle-Bleuez, Jean-Michel Vergnon, Fabien Forest

**Affiliations:** 1Department of Pathology, University Hospital of Saint Etienne, 42055 Saint Etienne, France; alice.court@etu.univ-st-etienne.fr (A.C.); david.laville@etu.univ-st-etienne.fr (D.L.); sami.dagher@etu.univ-st-etienne.fr (S.D.); vincent.grosjean@etu.univ-st-etienne.fr (V.G.); pierre.dal-col@chu-st-etienne.fr (P.D.-C.); violaine.yvorel@chu-st-etienne.fr (V.Y.); francois.casteillo@chu-st-etienne.fr (F.C.); 2Department of Molecular Biology of Solid Tumors, University Hospital of Saint Etienne, 42270 Saint Etienne, France; 3Department of Pneumology, University Hospital of Saint Etienne, 42270 Saint Etienne, France; sophie.bayle@chu-st-etienne.fr (S.B.-B.); j.michel.vergnon@chu-st-etienne.fr (J.-M.V.)

**Keywords:** adenocarcinoma, TTF-1, CK7, CK20, EGFR

## Abstract

Anti-CK7 and anti-CK20 immunohistochemistry is sometimes used to establish a diagnosis of primary lung cancer. We performed a retrospective study on the value of anti-CK7 and anti-CK20 immunohistochemistry in 359 biopsies of patients with suspected lung carcinoma in order to assess the usefulness of these antibodies in the evaluation of lung tumors in biopsies. Our results showed TTF-1 positivity in 73.3% of patients. *EGFR* mutations and *ALK* rearrangements were significantly different between TTF-1 positive and TTF-1 negative tumors (*p* < 0.001 and *p* = 0.023, respectively). Our results show a significant difference (*p* < 0.001) between TTF-1 positive and TTF-1 negative carcinomas with a median survival of 21.97 months (CI95% = 17.48–30.9 months) and 6.52 months (CI95% = 3.34–10.3 months), respectively. In the group of TTF-1 negative patients, anti-CK7 and CK20 immunohistochemistry was performed in 70 patients and showed CK7+/CK20- staining in 61 patients (87.1%), CK7-/CK20- in 4 patients (5.7%), CK7+/CK20+ in 3 patients (4.3%), and CK7-/CK20- in 2 patients (2.8%). No specific or molecular pattern was found in these groups of CK7/CK20 combinations. In total, this work brings arguments concerning the uselessness of anti-CK7/CK20 immunohistochemistry in the case of suspicion of primary lung cancer in biopsies.

## 1. Introduction

Lung cancer is a common tumor, unfortunately often diagnosed at the metastasis stage. The three main types of lung cancer include adenocarcinoma, squamous cell carcinoma, and small cell carcinoma. The diagnosis is often made on small specimens such as biopsies. Immunohistochemistry is a protein detection technique commonly used in pathology. This technique allows to detect the presence of certain proteins, but also to localize these proteins on the subcellular level [1]. Immunohistochemistry in diagnostic pathology is based on the reaction between an antigen, a primary antibody, and a secondary antibody [2]. Immunohistochemistry is the cornerstone of diagnosis for lung tumors [2]. Indeed, immunohistochemistry is important to better distinguish the histological type of lung cancer [2]. Prior to the advent of targeted therapies, there was little difference in treatment by histological type in the non-small cell carcinoma group. One of the most important issues for the pathologist was to distinguish small cell carcinoma from non-small cell carcinoma, which is often possible on morphology alone. With the introduction of targeted therapies in the treatment of lung cancer, it has become crucial to determine the histological type of these tumors. Indeed, targeted therapies have little or no indication in squamous cell carcinomas. The development of anti-p40 immunohistochemistry and its sensitivity close to 100% for the diagnosis of squamous cell carcinoma has allowed a better typing of lung carcinomas and a better separation of squamous cell carcinomas from “non-small cell non-squamous” carcinomas [3]. In parallel with the need to type and subtype lung carcinomas, complementary techniques for therapeutic purposes such as molecular biology have required changes in practice for pathologists with the sparing of tissue material [4]. Recommendations regarding tissue sparing were issued in 2011 and recommended to use minimal stains to diagnose non-small cell lung carcinoma [5].

When a patient presents with a lung tumor, clinically the probability that it is a metastasis of a cancer of another origin is high [6]. Primary lung tumors are less common than lung metastases in clinical practice [6]. Anti-CK7 and CK20 antibodies are important in suggesting the site of origin of the most common cancers. In pathological routine practice, anti-CK7 and CK20 immunohistochemistry may be requested for diagnostic guidance, especially for TTF-1 negative and p40 negative lung cancer [7]. However, all labeling combinations are possible in primary lung tumors, although the CK7+/CK20- profile is the most common [8]. For example, the CK7-/CK20+ combination may point to a colorectal origin. On the other hand, the CK7+/CK20- combination may be consistent with a pulmonary origin, but may also be found in thyroid, salivary gland, mammary, endometrial, ovarian, or mesothelial origins. Nevertheless, primary lung adenocarcinomas can be positive for CK20 especially in mucinous, colloid, and enteric subtypes [9]. Furthermore, CK7 tends to stain more often adenocarcinoma than squamous cell carcinoma, but cannot be used to discriminate adenocarcinoma from squamous cell carcinoma [10]. Squamous cell carcinomas on the other hand are mostly negative with the anti-CK20 antibody [11]. The CK7/CK20 pair has long been used in the diagnosis of lung tumors. Moreover, as the most frequent lung tumor is a metastasis of another cancer, it may be discussed to perform these immunohistochemical markers, which are the basis of important orientations in the case of carcinomas of unknown origin. TTF-1 is frequently expressed in 3/4 of lung adenocarcinomas and can be used as a tumor origin marker. ALK and ROS1 are immunohistochemical markers used to screen patients with rearrangements of these genes in routine practice. Our study was performed only in patients with suspected primary lung carcinoma. Nevertheless, the value of anti-CK7 and CK20 immunohistochemistry might be useful in the case of suspected metastasis.

Given that recommendations for tissue sparing are based on unproven expert opinion, in addition in current practice, the combination of anti-CK7/CK20 antibodies is sometimes requested, and we propose to investigate the diagnostic value of CK7 and CK20 immunohistochemistry in bronchial biopsies. As these recommendations are based on expert recommendations, and have not been proven on small specimens, we propose to evaluate the value of the combination of anti-CK7/CK20 immunohistochemistry on small specimens in a retrospective cohort study of patients with clinically suspected primary lung carcinoma. We propose to correlate the immunohistochemical data with molecular biological data and clinical follow-up. The purpose of our work is to support expert recommendations with robust real life clinical data.

## 2. Material and Methods

### 2.1. Patients

This study was approved by the ethics committee of University Hospital Center of Saint Etienne (IRBN112022/CHUSTE, Terres d’Ethique). Our work follows the recommendations of the European General Data Protection Regulation and all patients were informed about the study. All patients with bronchial or lung biopsy without clinical suspicion of lung metastasis from another cancer, with p40 negative carcinoma diagnosed between 2012 and 2020, were included. We collected the following clinical data: age at diagnosis, sex, localization of metastasis, date of last follow-up, or date of death. All patients were discussed in multidisciplinary case studies in our reference center. The site of origin was discussed between pathologists and oncologists in light of a complete evaluation of tumor extension.

### 2.2. Histopathologic Evaluation

All diagnoses were reviewed in light of the WHO 2021 classification [12]. Anti-CK7 and CK20 immunohistochemistry never exhausted the sample for further techniques such as immunohistochemistry or molecular biology. Automated immunohistochemistry was performed on 4 μm sections using the Omnis platform (Dako-Agilent, Courtaboeuf, France) according to the manufacturer’s instructions. Immunohistochemistry was performed on OMNIS (Dako-Agilent) using anti-TTF-1 antibodies (8G7G3/1, 1/50, Dako-Agilent). TTF-1 expression was considered positive if more than 5% of the tumor cells were nuclear stained. In the case of negativity, anti-CK7 (OV-TL 2/30, 1/600, Dako-Agilent) and CK20 (Ks20.8, 1/100, Dako-Agilent) were used. Mucin stain was used to confirm the adenocarcinoma in the case of negative TTF-1 and Napsin A (IP64, Leica) immunohistochemistry was perfomed in negative TTF-1.

ALK (D5F3, 1/100, Cell Signaling Technology, Danvers, MA, USA) and ROS1 (D4D6, 1/40 Cell Signaling Technology) antibodies were used for the screening of *ALK* and *ROS1* rearrangements. When positive or doubtful, another technique was used to confirm *ALK* or *ROS1* rearrangement.

### 2.3. Molecular Analysis

Molecular testing was performed as previously described [4,13]. *EGFR* (epidermal growth factor receptor) mutations were screened using a peptic nucleic acid (PNA) clamp detection kit (Entrogen detection kit, Entrogen Inc., Woddland Hill, CA, USA) or via next-generation sequencing (NGS) on the Personal Genome Machine (Thermofisher, Dardilly, France). *KRAS* (V-Ki-ras2 Kirsten Rat Sarcoma viral oncogene Homolog) mutations were screened via PNA clamp, Snapshot, or NGS. Exon 15 mutations of *BRAF* (v-Raf murine sarcoma viral oncogene homolog B) were analyzed using Snapshot or NGS. *ERBB2* (HER2) mutations of exon 20 were screened for via direct sequencing or NGS. When the tumour cell proportion was below the detection limit of techniques, the results were not considered [14].

Library preparation was carried out using amplicon technology with the Ion AmpliSeq ™ Colon and Lung V2 ready-to-use panel (Thermofisher), which amplifies genomic hot spot regions that are frequently mutated in human cancer genes. This panel consists of a single pool of primers and associated reagents used in multiplex PCR for the preparation of amplicon libraries for NGS and is designed to amplify 92 amplicons covering various tumour-associated mutations across 22 genes (*EGFR*, *ALK*, *ERBB2*, *ERBB4*, *FGFR1*, *FGFR2*, *FGFR3*, *MET*, *DDR2*, *KRAS*, *PIK3CA*, *BRAF*, *AKT1*, *PTEN*, *NRAS*, *MAP2K1*, *STK11*, *NOTCH1*, *CTNNB1*, *SMAD4*, *FBXW7*, *TP53*).

Clonal amplification and loading of sequencing chips were performed using the Ion Chef (Thermofisher), and sequencing was carried out on the Ion PGM (Thermofisher) with 500× minimum.

### 2.4. Statistical Analysis

Statistical analysis was performed using R software (version 3.4.1) with R Studio for Windows (version 1.0.143) [15]. The “survival” package was used for survival analysis (version 2.44-1.1) [16]. Fisher and χ^2^ tests were used to compare categorical variables when appropriate. Overall survival was calculated via the Kaplan–Meier method. The date of diagnosis to the date of death or censoring (if patients were alive at the time of last follow-up) was used to calculate overall survival.

## 3. Results

### 3.1. Patient Characteristics

Three hundred and fifty-nine patients were included in this study, suspected clinically and on imaging to be primary lung tumors. The mean age at diagnosis was 65.5 ± 11.1 years. There were 225 men (62.7%) and 134 women (37.3%). The final diagnosis was adenocarcinoma in 302 patients (84.1%), non-small cell carcinoma in 54 patients (15%), carcinomatous lymphangitis in 2 patients (0.6%), and suspicion of sarcomatoid carcinoma in 1 patient (0.3%).

### 3.2. TTF-1 Negative versus TTF-1 Positive Groups

TTF-1 was positive in 263 (73.3%) patients and negative in 91 (25.3%), and TTF-1 was not performed in 5 patients because the tumor material was not sufficient in order to favor molecular biology testing.

We compared the dissemination at the main metastatic sites between TTF-1 positive and TTF-1 negative carcinomas (Table 1); our results show a higher proportion of adrenal metastases in TTF-1 negative patients (*p* = 0.039). There was no statistically significant difference in the proportion of lung, pleural, bone, brain, and liver metastases between the TTF-1 positive and TTF-1 negative groups. There was a trend towards a higher proportion of pleural metastases in the TTF-1 positive group (*p* = 0.089). The most frequent metastatic sites were the lung in 70 (21.8%) patients, the lymph nodes in 71 (22.1%) patients, the pleura in 33 (10.3%) patients, the adrenals in 66 (20.6%) patients, the brain in 113 (35.2%) patients, the liver in 56 (17.4%) patients, and other sites in 26 (8.1%) patients. Two hundred and thirty-nine patients (74.4%) had no metastasis at diagnosis.

We compared the molecular profile used routinely between TTF-1 positive and TTF-1 negative carcinomas (Table 2). *EGFR* mutations and *ALK* rearrangements were significantly different between TTF-1 positive and TTF-1 negative tumors (*p* < 0.001 and *p* = 0.023 respectively). *EGFR* mutations were present in 16.3% of TTF-1 positive patients, while they were present in only 1.1% of TTF-1 negative patients. *ALK* rearrangements were found in 5.3% of TTF-1 positive patients, whereas no *ALK*-rearranged patient was found among TTF-1 negative patients. There was no significant difference between the two groups for *KRAS* mutations, *BRAF*, and *ROS1* rearrangements.

Finally, we compared the overall survival between TTF-1 positive and TTF-1 negative carcinomas (Figure 1). Our results show a significant difference (*p* < 0.001) between TTF-1 positive and TTF-1 negative carcinomas with a median survival of 21.97 months (CI95% = 17.48–30.9 months) and 6.52 months (CI95% = 3.34–10.3 months), respectively.

### 3.3. Anti-CK7 and CK20 Immunohistochemistry

As TTF-1 negativity did not provide an argument for primary pulmonary origin, we performed anti-CK7 and CK20 immunohistochemistry in this group. Napsin A was performed in TTF-1 negative patients, 3 of them were Napsin A positive. In the group of TTF-1 negative patients, anti-CK7 and CK20 immunohistochemistry was performed in 70 patients and showed CK7+/CK20- staining in 61 patients (87.1%), CK7-/CK20- in 4 patients (5.7%), CK7+/CK20+ in 3 patients (4.3%), and CK7-/CK20- in 2 patients (2.8%) (Figure 2). CK7 and CK20 was not performed for 21 patients in order to spare tumor tissue. There was no significant difference in the metastatic dissemination of these tumors regardless of their CK7/CK20 status (Table 3). There was a trend towards a significant difference (*p* = 0.078) for liver metastases, but with very small numbers in these subgroups (n = 2 to 4). There was no significant difference in *EGFR*, *KRAS*, *BRAF*, *ALK*, and ROS1 status according to CK7/CK20 status (Table 4). The CK7+/CK20- group had 37.7% of KRAS mutations. Survival analysis in these small subgroups was not performed because of the small numbers. Among the TTF-1 negative and KRAS mutated carcinomas, 14 of them had the p.G12C mutation in exon 2.

## 4. Discussion

Our work on this retrospective cohort demonstrates that anti-CK7/CK20 immunohistochemistry in biopsies in patients with suspected TTF-1 and p40 negative primary lung cancer is of little value. The use of CK7/CK20 immunohistochemistry in this setting did not demonstrate with certainty that the carcinoma was of another origin. Moreover, the different CK7/CK20 profiles do not allow to identify a particular population either in terms of metastatic evolution or in terms of molecular profile. This work brings additional arguments demonstrating the uselessness of the use of the anti-CK7/CK20 couple in this context. The strength of our work is that the pathological data are correlated with the clinical follow-up data and molecular data. Our work was performed in a particular population where the pathological examination is performed considering the clinical context. Indeed, when the biopsy was performed to authenticate a metastatic progression of a known cancer, it was not included in our study. In this specific case, which differs from the conditions of inclusion of our work, the anti-CK7/CK20 immunohistochemistry can be an element of orientation in certain cases [7]. It is thus crucial to examine the biopsies with all the appropriate clinical information, thus making better use of the tumor material, sparing it for complementary techniques such as molecular biology, and answering the questions asked by the pathology exams prescribers as accurately as possible.

In contrast, the distinction between TTF-1 positive and TTF-1 negative non-squamous carcinomas seems to be clinically relevant with a more favorable prognosis in the case of TTF-1 positivity. In contrast, the distinction between TTF-1 positive and TTF-1 negative non-squamous carcinomas is more relevant. This distinction seems to be clinically relevant with a more favorable prognosis in the case of TTF-1 positivity. Moreover, at the molecular level, the probability of finding an *EGFR* mutation or an *ALK* rearrangement is higher in the case of TTF-1 positivity. These data correlate with what has already been described in the literature where TTF-1 has been shown to be a better prognostic factor in adenocarcinomas of all stages and where TTF-1 positivity increases the probability of finding *EGFR* or *ALK* mutations [17,18,19,20,21,22,23]. TTF-1 is a marker present in more than 60% of primary lung adenocarcinomas, and is rarely expressed in adenocarcinomas of other origin [2,17,24]. TTF-1 is a transcription factor involved in morphogenesis, differentiation, and surfactant production by normal pneumocytes [25]. The different frequency of *EGFR* and *ALK* subtypes between TTF-1 positive and TTF-1 negative subtypes might be linked to the fact that TTF-1 negative tumors in biopsies represent a group enriched in rare adenocarcinomas subtypes. These subtypes are mostly represented by invasive mucinous adenocarcinoma representing roughly 10% of lung tumors and are often *EGFR* wild type and TTF-1 negative [2]. It thus seems logical to distinguish TTF-1 positive non-squamous carcinomas, which would be more differentiated, from TTF-1 negative ones, which would be less differentiated. Finally, it seems that, among TTF-1 negative carcinomas, the proportion of rare carcinomas such as carcinomas muted for *SMARCA4* would be more frequent [26,27]. The exact place of *SMARCA4* testing as far as there is a specific antibody for this diagnosis is not resolved in the setting of the diagnosis on small biopsy samples [26,27]. Furthermore, because of the recent description of *SMARCA4* deficient tumors, the therapy is not as well-known compared with lung adenocarcinoma [28]. The subtyping of rare lung carcinoma such as enteric carcinoma and sarcomatoid carcinoma, especially on bronchial biopsies, is difficult or not feasible, and there is often little or no difference in the treatment between these subtypes [29].

Cytokeratins belong to the family of intermediate filaments. Cytokeratin 7 marks the simple epithelia, while cytokeratin 20 marks the simple epithelia of the digestive tract and Merkel cells [11]. The anti-CK7/CK20 pair may be of diagnostic interest in the context of adenocarcinoma metastasis [10]. Furthermore, primary lung adenocarcinomas can have any combination, although the CK7 positive and CK20 negative combination is the most common [7]. Among TTF-1 negative carcinomas, the proportion of carcinomas with a profile other than CK7+/CK20- remains rare and concerns only 12.8% of cases in our series. The possible interest of using cytokeratins is to prove that it is an epithelial tumor. The vast majority of non-epithelial tumors do not express cytokeratins unlike carcinomas [10]. However, the vast majority of pulmonary malignancies are carcinomas, and it is morphologically possible to suspect that it is an epithelial tumor.

The limitations of our work are inherent to those of retrospective work on existing data with a proportion of missing or unrealized data. Moreover, this is a single-center study in a university hospital of reference in which clinical and pathological collaboration is important. Indeed, the pathologists systematically have the patient’s medical history, the indication for endoscopy, and the endoscopic data. However, clinical information is not always provided to the pathologist in less specialized centers. Another limitation of our work is that we are interested in a particular subgroup concerning TTF-1 negative carcinomas. This induces de facto a low number of patients in the subgroup of TTF-1 negative patients and limits the relevance of the statistical analysis. Biopsies are fixed in formalin immediately and there is no delay in fixation, as may be the case for surgical specimens. However, the different pre-analytical conditions can induce differences in marking, which is not the case in this work [30].

The diagnosis of lung cancer has completely changed in recent years. Indeed, pathological diagnosis has had to adapt to molecular techniques to save as much tissue as possible for these techniques. Indeed, even with new generation sequencing capable of sequencing several genes in parallel, it is necessary to have a maximum amount of tumor tissue to limit the risk of false negative or exhausted samples in molecular biology [4,31]. Moreover, concerning immunohistochemical techniques, it is recommended to privilege as much as possible techniques having a therapeutic impact, such as anti-PD-L1 immunohistochemistry [32,33]. Molecular techniques are of major importance, for example, concerning the *EGFR* gene, which is mutated in 10–15% of non-Asian patients and in 40–60% of Asian patients with lung adenocarcinoma. *EGFR* allows to determine the therapy; that is, metastatic patients with activating mutations of *EGFR* can be treated with tyrosine kinase inhibitors, while negative patients will not benefit from these specific therapies. However, the prognosis of *EGFR*-mutated metastatic patients treated with tyrosine kinase inhibitors is much better than that of the *EGFR*-wild type group [34]. It is thus more logical to focus on the most clinically impactful techniques. A biopsy is a small sample on which it is necessary to make a diagnosis and determine which molecular or immunohistochemical markers are present. These samples can be quickly exhausted by successive complementary techniques. Furthermore, there is no single technique that can provide all the data needed for treatment. Techniques to determine markers for therapy are multiple and include immunohistochemistry for ALK, ROS1, and PD-L1, but also sequencing ideally of DNA as well as of RNA for gene rearrangements [35]. Pathologists need to be aware of therapeutic developments to better support diagnostic and therapeutic techniques and better manage tissue material. Artificial intelligence techniques, without the need for additional techniques using tissue, could be of interest to better classify these tumours. Their use in routine practice is still in the early stages, but could be of interest in the future [36].

Overall, this work brings arguments concerning the uselessness of anti-CK7/CK20 immunohistochemistry in most cases of suspicion of primary lung cancer in biopsies in routine clinical practice. This pair of antibodies does not allow the identification of relevant subgroups in terms of molecular group, metastasis, or prognosis in our work. The use of this pair of antibodies could suggest a primary tumor of another origin and lead to a lengthening of the management time and the realization of unnecessary clinical complementary examinations. In addition, the use of antibodies leads to tissue consumption, whereas it is essential to spare tumor tissue for the realization of complementary examinations that could lead to the loss of tumor tissue. In the era of personalized therapies, it is thus legitimate that recommendations privilege tissue sparing rather than the realization of complementary techniques. The use of immunohistochemistry should be done sparingly. No diagnosis of cancer origin can be made on immunohistochemistry alone. The diagnosis of the origin of a cancer must imperatively integrate the associated clinical and radiological data. However, our study was performed in patients with suspected lung cancer. It is not applicable in patients suspected of having metastasized from another cancer. Our results are thus applicable in a selected population and do not account for situations that are sometimes particular. Our results support the international recommendations that should be applied according to individual clinical situations.

To conclude, in the case of suspicion of primary lung carcinoma, we show that the use of anti-CK7 and CK20 antibodies does not help the diagnosis, and our work supports the recommendations made.

## Figures and Tables

**Figure 1 diagnostics-12-01589-f001:**
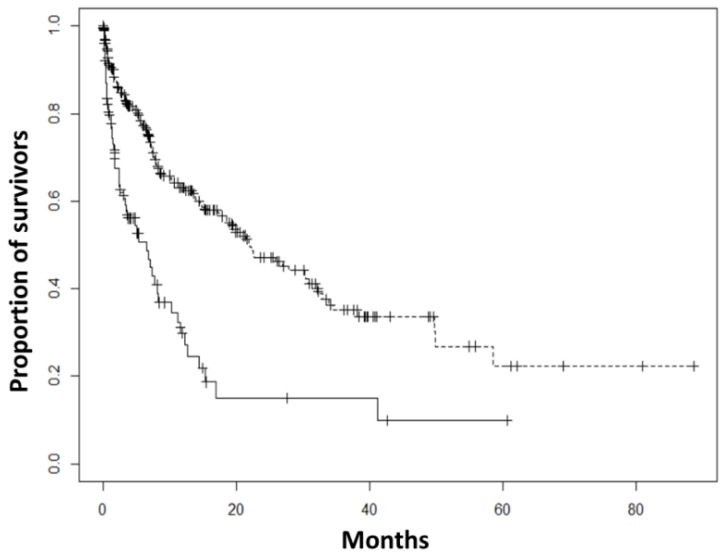
Overall survival according to TTF-1 status. Dash: TTF-1 positive carcinomas. Solid line: TTF-1 negative carcinomas.

**Figure 2 diagnostics-12-01589-f002:**
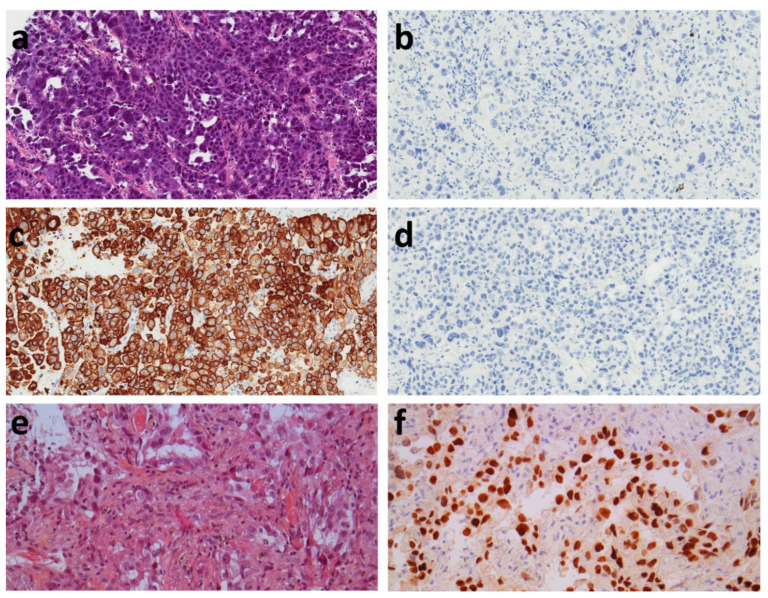
Illustrative microphotograph of TTF-1 negative, CK7 positive, and CK20 negative lung carcinoma. (**a**) Hematoxylin Eosin, ×200, showing a lung carcinoma without neuroendocrine morphology. (**b**) TTF-1 immunohistochemistry, ×200, tumor cells are not stained. (**c**) CK7 immunohistochemistry, ×200, showing a diffuse staining of tumor cells. (**d**) CK20 immunohistochemistry, ×200, tumor cells are not stained. (**e**) Hematoxylin Eosin, ×200, showing a lung carcinoma without neuroendocrine morphology. (**f**) TTF-1 immunohistochemistry, ×200, nuclear staining of tumor cells.

**Table 1 diagnostics-12-01589-t001:** Metastatic sites according to TTF-1 status.

Metastatic Site	TTF-1 Positive, n (%)	TTF-1 Negative, n (%)	*p*
**n (%)**	240 (66.5)	81 (22.4)	
**Lung**			0.716
Yes	51 (19.4)	19 (20.9)	
No	189 (71.8)	62 (68.1)	
**Lymph nodes**			0.206
Yes	49 (18.6)	22 (24.2)	
No	191 (72.6)	59 (64.8)	
**Pleura**			0.089 *
Yes	29 (11)	4 (4.4)	
No	211 (80.2)	77 (84.6)	
**Bone**			0.634
Yes	84 (31.9)	26 (28.6)	
No	156 (59.3)	55 (60.4)	
**Adrenals**			**0.039**
Yes	40 (15.2)	22 (24.2)	
No	200 (76)	59 (64.8)	
**Brain**			0.224
Yes	89 (33.8)	24 (26.4)	
No	151 (57.4)	57 (62.6)	
**Liver**			0.965
Yes	42 (16)	14 (15.4)	
No	198 (75.3)	67 (73.6)	
**Other metastatic site**	15 (5.7)	11 (12.1)	
**No metastasis**	179 (68.6)	60 (65.9)	
**Data not available**	23 (8.8)	10 (11)	

*: Fisher exact test.

**Table 2 diagnostics-12-01589-t002:** Molecular alterations between TTF-1 positive and TTF-1 negative groups.

	TTF-1 Positive, n (%)	TTF-1 Negative, n (%)	*p*
** *EGFR* **			**<0.001**
Mutated	43 (16.3)	1 (1.1)	
Wild type	192 (73)	79 (86.8)	
Not performed	28 (10.6)	11 (12.1)	
** *KRAS* **			
Mutated	75 (28.5)	33 (36.3)	0.148
Wild type	157 (59.7)	47 (51.6)	
Not performed	31 (11.8)	11 (12.1)	
** *BRAF* **			0.65
Mutated	11 (4.2)	3 (3.3)	
Wild type	196 (74.5)	72 (79.1)	
Not performed	56 (21.3)	16 (17.6)	
** *ALK* **			**0.023**
Rearranged	14 (5.3)	0 (0)	
Not rearranged	214 (81.4)	80 (87.9)	
Not performed	35 (13.3)	11 (12.1)	
** *ROS1* **			0.106
Rearranged	6 (2.3)	0 (0)	
Not rearranged	150 (57)	66 (72.5)	
Not performed	107 (40.7)	25 (27.5)	

**Table 3 diagnostics-12-01589-t003:** Metastatic sites according to CK7/CK20 status in TTF-1 negative carcinomas.

Metastatic Site for TTF-1 Negative	CK7+/CK20-, n (%)	CK7-/CK20-, n (%)	CK7+/CK20+, n (%)	CK7-/CK20+, n (%)	*p*
**n (%)**	61 (87.1)	4 (5.7)	3 (4.3)	2 (2.8)	
**Lung**					0.771 *
Yes	10 (16.4)	0 (0)	1 (33.3)	0 (0)	
No	44 (72.1)	2 (50)	2 (66.7)	2 (100)	
**Lymph nodes**					0.892 *
Yes	16 (26.2)	0 (0)	1 (33.3)	1 (50)	
No	38 (62.3)	2 (50)	2 (66.7)	1 (50)	
**Pleura**					0.191 *
Yes	2 (3.3)	0 (0)	0 (0)	1 (50)	
No	52 (85.2)	2 (50)	3 (100)	1 (50)	
**Bone**					0.672 *
Yes	23 (37.7)	0 (0)	1 (33.3)	0 (0)	
No	31 (50.8)	2 (50)	2 (66.7)	2 (100)	
**Adrenals**					0.258 *
Yes	14 (22.9)	0 (0)	2 (66.7)	1 (50)	
No	40 (65.6)	2 (50)	1 (33.3)	1 (50)	
**Brain**					0.621 *
Yes	19 (31.1)	0 (0)	0 (0)	0 (0)	
No	38 (62.3)	2 (50)	3 (100)	2 (100)	
**Liver**					0.078
Yes	7 (11.5)	0 (0)	2 (66.7)	0 (0)	
No	47 (77)	2 (50)	1 (33.3)	0 (0)	
**Other metastatic site**	9 (14.7)	0 (0)	0 (0)	0	
**No metastasis**	35 (57.4)	2 (50)	1 (33.3)	2 (100)	
**Data not available**	7 (11.5)	2 (50)	0 (0)	0 (0)	

*: Fisher exact test.

**Table 4 diagnostics-12-01589-t004:** Molecular alterations between CK7/CK20 groups in TTF-1 negative carcinomas.

	CK7+/CK20-, n (%)	CK7-/CK20-, n (%)	CK7+/CK20+, n (%)	CK7-/CK20+, n (%)	*p*
** *EGFR* **	61	4	3	2	0.081
Mutated	0 (0)	0 (0)	1 (33.3)	0 (0)	
Wild type	53 (86.9)	4 (100)	2 (66.7)	2 (100)	
Not performed	8 (11.5)	0 (0)	0 (0)	0 (0)	
** *KRAS* **					0.118
Mutated	23 (37.7)	0 (0)	0 (0)	0 (0)	
Wild type	30 (49.2)	4 (100)	3 (100)	2 (100)	
Not performed	8 (13.1)	0 (0)	0 (0)	0 (0)	
** *BRAF* **					0.397
Mutated	2 (3.3)	1 (25)	0 (0)	0 (0)	
Wild type	48 (78.7)	3 (75)	3 (100)	2 (100)	
Not performed	11 (18)	0 (0)	0 (0)	0 (0)	
** *ALK* **					1
Rearranged	0 (0)	0 (0)	0 (0)	0 (0)	
Not rearranged	53 (86.9)	4 (100)	2 (66.7)	2 (100)	
Not performed	8 (13.1)	0 (0)	1 (33.3)	0 (0)	
** *ROS1* **					1
Rearranged	0 (0)	0 (0)	0 (0)	0 (0)	
Not rearranged	43 (70.5)	4 (100)	0 (0)	2 (100)	
Not performed	18 (29.5)	0 (0)	3 (100)	0 (0)	

## Data Availability

Data available on request.

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
