# Peer review of "Anti-CK7/CK20 Immunohistochemistry Did Not Associate with the Metastatic Site in TTF-1-Negative Lung Cancer"

_diagnostics, 2022, doi:10.3390/diagnostics12071589_

Round 1

Reviewer 1 Report

The research paper of Allegra Ferrari et al. titled " Anti-CK7 and CK20 Immunohistochemistry is Unnecessary in the Diagnosis of Lung Tumors" is an informative study that aims to investigate the diagnostic value of CK7 and CK20 immunohistochemistry on bronchial biopsies. However, there are several minor issues with this study.

  • Introduction:
  1. The first paragraph is extremely general and it is suggested that the authors provide more comprehensive information. Additionally, the authors could give additional details about lung cancer.
  2. In the second paragraph, the authors are advised to provide details on TTF-1, ALK, and ROS1, highlighting the reason(s) for which they chose these molecules.
  • Results:
  1. A table with the clinicopathological characteristics of patients should be included.
  2. Figure 1 is not cited in the text, please fix this issue.
  • Discussion:
  1. Figure 2 is not referenced in the text, please fix this issue. Moreover, this figure is suitable for the Results section.
  2. The Discussion is very lengthy and includes too many details. The Authors are advised to be more careful in the content and more focused on the most important parts of their study. Additional commentary could be added to the Discussion, regarding the importance of the findings and their potential utilization in standard clinical practice. Lastly, the authors could propose alternative techniques that could replace Immunohistochemistry and could help in the diagnosis in case of suspicion of primary lung carcinoma.
  • Proofreading of the manuscript is needed.

Author Response

The research paper of Allegra Ferrari et al. titled " Anti-CK7 and CK20 Immunohistochemistry is Unnecessary in the Diagnosis of Lung Tumors" is an informative study that aims to investigate the diagnostic value of CK7 and CK20 immunohistochemistry on bronchial biopsies. However, there are several minor issues with this study.

  • Introduction:
  1. The first paragraph is extremely general and it is suggested that the authors provide more comprehensive information. Additionally, the authors could give additional details about lung cancer. -> We have added the following sentences : « Lung cancer is a common tumor, unfortunately often diagnosed at the metastasis stage. The three main types of lung cancer include adenocarcinoma, squamous cell carcinoma and small cell carcinoma. The diagnosis is often made on small specimens such as biopsies. »
  2. In the second paragraph, the authors are advised to provide details on TTF-1, ALK, and ROS1, highlighting the reason(s) for which they chose these molecules. -> We have added the following sentences : « TTF-1 is frequently expressed in 3/4 of lung adenocarcinomas and can be used as a tumor origin marker. ALK and ROS1 are immunohistochemical markers used to screen patients with rearrangements of these genes in routine. »
  • Results:
  1. A table with the clinicopathological characteristics of patients should be included. -> We did no wish to include a table with the clinicopathological characteristics because the first paragraph of the results summarizes the clinicopathological characteristics and because we found it was redundant with other tables.
  2. Figure 1 is not cited in the text, please fix this issue. -> We apologize for this issue, we have cited Fig.1
  • Discussion:
  1. Figure 2 is not referenced in the text, please fix this issue. Moreover, this figure is suitable for the Results section. -> We apologize for this issue, we have cited Fig.2

  1. The Discussion is very lengthy and includes too many details. The Authors are advised to be more careful in the content and more focused on the most important parts of their study. Additional commentary could be added to the Discussion, regarding the importance of the findings and their potential utilization in standard clinical practice. Lastly, the authors could propose alternative techniques that could replace Immunohistochemistry and could help in the diagnosis in case of suspicion of primary lung carcinoma. -> The Editor requested that we increase our manuscript to more than 3000 words, so we had to lengthen the initial discussion. We have added the following sentences in the discussion section : “Artificial intelligence techniques, without the need for additional techniques using tissue, could be of interest to better classify these tumours. Their use in routine practice is still early, but could be of interest in the future [36].”
  • Proofreading of the manuscript is needed. -> We have carefully proofread the manuscript again and apologize for the inconvenience.

Reviewer 2 Report

This manuscript may be a beneficial examination for useful in the effective use of the bronchoscope specimen of "primary suspected" lung cancer. Indeed, I am interested in immunostaining and relations of the mutation in the gene. It is very important for clinitian that a specimen left to examine a gene. However, a problem is still left unfinished about the evaluation with the treatment response.
It is useful and I think that it was able to be a valuable report by major revision. Here is a summary of steps that I would recommend:

Major revision:

1. P6 L180 "There was a trend towards a significant difference (p=0.07) for EGFR, but with very small numbers in these subgroups (n=2 to 4)." 
The "p = 0.07" value was not able to find a mention to me from a table 4.
Although authers mentioned very small numbers, I think that you should not mention the tendency of the statistical significant difference in this EGFR group.
There was not the statistical significant difference, but I am interested in only the CK7+/CK20- group had 37.7% of KRAS mutations.

Minor revision:

1. P6 Tabel 4. Please revise CK7-/CK20- for CK7-/CK20+.
2. P7 L204-207 We see it so that the same sentence repeats to me. Please revise it.

Author Response

This manuscript may be a beneficial examination for useful in the effective use of the bronchoscope specimen of "primary suspected" lung cancer. Indeed, I am interested in immunostaining and relations of the mutation in the gene. It is very important for clinitian that a specimen left to examine a gene. However, a problem is still left unfinished about the evaluation with the treatment response.
It is useful and I think that it was able to be a valuable report by major revision. Here is a summary of steps that I would recommend:

Major revision:

  1. P6 L180 "There was a trend towards a significant difference (p=0.07) for EGFR, but with very small numbers in these subgroups (n=2 to 4)." 
    The "p = 0.07" value was not able to find a mention to me from a table 4.
    Although authers mentioned very small numbers, I think that you should not mention the tendency of the statistical significant difference in this EGFR group. -> We agree with reviewer 2 and have deleted this sentence.
    There was not the statistical significant difference, but I am interested in only the CK7+/CK20- group had 37.7% of KRAS mutations.-> We have underlined this information in the text.

Minor revision:

  1. P6 Tabel 4. Please revise CK7-/CK20- for CK7-/CK20+.-> We apologize for this mistake and have corrected ths.
    2. P7 L204-207 We see it so that the same sentence repeats to me. Please revise it.-> We apologize for this and have modified the sentence.

Reviewer 3 Report

Dear authors,

The manuscript diagnostics-1682151 submitted to Diagnostics as an original research article, is a retrospective diagnostic accuracy study investigating the potential of anti-CK7/CK20 immunohistochemistry to predict i) the metastatic sites and ii) the survival in TTF-1 (-) lung cancers.

The authors conclude that anti-CK7/CK20 was not suggestive of any particular metastatic site in their studied population, whereas they avoid making conclusive statements on survival due to low statistical power.

The study makes a very relevant clinical point to the aims and scopes of Diagnostics, has a structured design, applies proper statistics, and has a high reading interest.

C001 - As a major point, the reviewer would like to mention that the main statement and the title, especially the utilization of the word “unnecessary”, “uselessness” and so on…) are maybe too “aggressive”, risky, and unprofessional, and should be avoided. As a plausible alternative title, the authors might consider something more directly justified by the results, such as “ anti-CK7/CK20 immunohistochemistry did not associate with the metastatic site in TTF-1-negative lung cancer”.

C002 – The manuscript will benefit from professional language proof editing.

Best regards,

Author Response

Dear authors,

The manuscript diagnostics-1682151 submitted to Diagnostics as an original research article, is a retrospective diagnostic accuracy study investigating the potential of anti-CK7/CK20 immunohistochemistry to predict i) the metastatic sites and ii) the survival in TTF-1 (-) lung cancers.

The authors conclude that anti-CK7/CK20 was not suggestive of any particular metastatic site in their studied population, whereas they avoid making conclusive statements on survival due to low statistical power.

The study makes a very relevant clinical point to the aims and scopes of Diagnostics, has a structured design, applies proper statistics, and has a high reading interest.

C001 - As a major point, the reviewer would like to mention that the main statement and the title, especially the utilization of the word “unnecessary”, “uselessness” and so on…) are maybe too “aggressive”, risky, and unprofessional, and should be avoided. As a plausible alternative title, the authors might consider something more directly justified by the results, such as “ anti-CK7/CK20 immunohistochemistry did not associate with the metastatic site in TTF-1-negative lung cancer”.-> We agree with reviewer 3, we have modified the title as suggested.

C002 – The manuscript will benefit from professional language proof editing. -> We have read and edited the manuscript. We apologize for the inconvenience.

Best regards,

Reviewer 4 Report

The authors described the use of CK7/CK20 in the diagnosis of lung cancer and concluded that CK7/CK20 is not useful. The study compared TTF1 positive and negative cases on their molecular mutation profile and metastatic sites.

It is generally well written and represented. However, the approach to answering the research question is not proper.

What is the significant of comparing molecular mutations between TTF1 positive and negative cases?

How do you confirm it is an adenocarcinoma when the TTF1 is negative? Was Napsin A performed? For TTF1 negative cases, performing Napsin A immunohistochemistry is a must.

EGFR mutation is more common in lung adenocarcinoma, is it possible that the TTF1 negative tumours were actually other types of lung cancer? If not, how did you confirm that?

WHO classification of tumour has recommended not to use CK7 and CK20 as biomarkers for lung cancer.

In Table 3, it is not clear, what do you mean metastatic site (lung)? Again, I don’t see the important of comparing different CK7/CK20 expression for TTF-1 negative tumour with metastatic site. CK7 and CK20 are epithelial related markers that are mainly used to determine the site of origin of a tumour/ tissue. Why would it be different in a same tumour with and without metastasis?

A figure of TTF1 positive tumour should also be included.

The conclusion of CK7 and CK20 is not useful for the diagnosis of primary lung carcinoma is not support by the current methodology. The approach is not correct. In order to answer this question, you will need to compare the sensitivity and specificity of TTF1 with CK7/CK20 on the lung carcinoma. In addition, a comparison of TTF1 and CK7/CK20 ability to distinguish metastatic tumour to the lung.

Comparing between adenocarcinoma and squamous cell carcinoma is not an issue, with TTF1, Napsin A, P63 and P40, it should be able to differentiate them.

Comparing between adenocarcinoma and small cell carcinoma is generally not an issue too, as neuroendocrine markers will help.

The main problem arises when there is a metastatic adenocarcinoma to the lung. A panel of biomarkers is needed to aid in diagnosis.

The discussion is not focus; how molecular mutation relate to the diagnosis. Further discussion is needed to describe the current update on biomarkers of lung cancer. In addition, further discussion on the findings of significant difference in EGFR and ALK mutations between TTF1 positive and negative tumours. 

Author Response

The authors described the use of CK7/CK20 in the diagnosis of lung cancer and concluded that CK7/CK20 is not useful. The study compared TTF1 positive and negative cases on their molecular mutation profile and metastatic sites.

It is generally well written and represented. However, the approach to answering the research question is not proper.

What is the significant of comparing molecular mutations between TTF1 positive and negative cases? -> The aim is to show that the group of TTF-1 positive tumours is different from the group of TTF-1 negative tumours. Furthermore, this approach is supported not only by the molecular differences but also by the prognostic differences. This is discussed in the following sentences in the discussion section : “Moreover, at the molecular level, the probability of finding an EGFR mutation or an ALK rearrangement is higher in case of TTF-1 positivity. These data correlate with what has already been described in the literature where TTF-1 has been shown to be a better prognostic factor in adenocarcinomas of all stages and where TTF-1 positivity increases the probability of finding EGFR or ALK mutations[17–23]. TTF-1 is a marker present in more than 60% of primary lung adenocarcinomas, and is rarely expressed in adenocarcinomas of other origin [2,17,24]. TTF-1 is a transcription factor involved in morphogenesis, differentiation and surfactant production by normal pneumocytes[25]. It therefore seems logical to distinguish TTF-1 positive non-squamous carcinomas, which would be more differentiated, from TTF-1 negative ones, which would be less differentiated.”

How do you confirm it is an adenocarcinoma when the TTF1 is negative? Was Napsin A performed? For TTF1 negative cases, performing Napsin A immunohistochemistry is a must.-> In the material and methods section, we have stated that “All diagnoses were reviewed in light of the WHO 2021 classification”. We have added the following sentence : “Mucin stain was used to confirm the adenocarcinoma in case of negative TTF-1”. We do not use routinely Napsin A in our lab because of its lower sensitivity and the same specificity than TTF-1.

EGFR mutation is more common in lung adenocarcinoma, is it possible that the TTF1 negative tumours were actually other types of lung cancer? If not, how did you confirm that? -> All tumors were adenocarcinoma as confirmed with mucin stain. We have added the following sentence : “All patients were discussed in multidisciplinary case studies in our reference center. The site of origin was discussed between pathologists, oncologist in light of complete evaluation of tumor extension.”

WHO classification of tumour has recommended not to use CK7 and CK20 as biomarkers for lung cancer. -> We agree with this, as stated in the introduction. Nevertheless, this recommendation is based on expert opinion and have not been proved. The aim of our work is to strengthen this recommendation.

In Table 3, it is not clear, what do you mean metastatic site (lung)? Again, I don’t see the important of comparing different CK7/CK20 expression for TTF-1 negative tumour with metastatic site. CK7 and CK20 are epithelial related markers that are mainly used to determine the site of origin of a tumour/ tissue. Why would it be different in a same tumour with and without metastasis? -> The lung is the most frequent metastatic site for primary tumors of the lung. The site of metastasis is related to its origin : lung tumor often metastasize in lung or pleura, colon adenocarcinoma frequently metastasize to liver…etc. The aim of this table is to show that metastatic dissemination is the same in all CK7/CK20 profiles.

A figure of TTF1 positive tumour should also be included.-> We have added an image of a TTF-1 positive tumor.

The conclusion of CK7 and CK20 is not useful for the diagnosis of primary lung carcinoma is not support by the current methodology. The approach is not correct. In order to answer this question, you will need to compare the sensitivity and specificity of TTF1 with CK7/CK20 on the lung carcinoma. In addition, a comparison of TTF1 and CK7/CK20 ability to distinguish metastatic tumour to the lung. -> No work has a perfect methodology, nevertheless, our work is based on a practical point of view. We disagree with this comment, as suggested by other reviewers who found that the methology was correct. We have discussed the limitations of our work in the discussion section.

Comparing between adenocarcinoma and squamous cell carcinoma is not an issue, with TTF1, Napsin A, P63 and P40, it should be able to differentiate them.Comparing between adenocarcinoma and small cell carcinoma is generally not an issue too, as neuroendocrine markers will help.The main problem arises when there is a metastatic adenocarcinoma to the lung. A panel of biomarkers is needed to aid in diagnosis.-> We agree with this comment, nevertheless, our work is not based on obviously metastatic lung adenocarcinoma to the lung, but on biopsies of tumor suspected clinically and on imaging to be lung primary tumors. We have added this in the results section.

The discussion is not focus; how molecular mutation relate to the diagnosis. Further discussion is needed to describe the current update on biomarkers of lung cancer. In addition, further discussion on the findings of significant difference in EGFR and ALK mutations between TTF1 positive and negative tumours. -> We have added the following sentences in the discussion section : “The different frequency of EGFR and ALK subtypes between TTF-1 positive and TTF-1 negative subtypes might be linked to the fact that TTF-1 negative tumors on biopsies represent a group enriched in rare adenocarcinomas subtypes. These subtypes are mostly represented by invasive mucinous adenocarcinoma representing roughly 10% of lung tumors and is often EGFR wild type and TTF-1 negative [2]. »

Round 2

Reviewer 2 Report

This manuscript may be a beneficial examination for useful in the effective use of the bronchoscope specimen of "primary suspected" lung cancer and immunostaining and relations of the mutation in the gene.

Thank you very much for your revision. I think that it became a more attractive article.

Now we can use KRAS G12C inhibitor, we  want to find KRAS mutation positively from TTF-1 negative patient.

Author Response

We thank reviewer #2 for his comments, we have added the following information in the result section, "Among the TTF-1 negative and KRAS mutated carcinomas, 14 of them had the p.G12C mutation in exon 2."

Reviewer 4 Report

The use of mucin instead of Napsin A is unacceptable as mucin is not sensitive and most tumour loses its ability to produce mucin. Here is a recent publication in Science Report stating that mucin is much inferior in comparison to TTF1 and Napsin A. 

Reference: Micke P, Botling J, Mattsson JSM, Planck M, Tran L, Vidarsdottir H, Nodin B, Jirström K, Brunnström H. Mucin staining is of limited value in addition to basic immunohistochemical analyses in the diagnostics of non-small cell lung cancer. Sci Rep. 2019 Feb 4;9(1):1319. 

The TTF1 negative group still need to a Napsin A for confirmation. Otherwise this study can be misleading. 

This statement below need confirmation that the TTF-1 tumour is definitely lung adenocarcinoma. There is no basis so far to use molecular mutation as a modality to introduce a new group.

"The different frequency of EGFR and ALK subtypes between TTF-1 positive and TTF-1 negative subtypes might be linked to the fact that TTF-1 negative tumors on biopsies represent a group enriched in rare adenocarcinomas subtypes. These subtypes are mostly represented by invasive mucinous adenocarcinoma representing roughly 10% of lung tumors and is often EGFR wild type and TTF-1 negative [2]."

Author Response

We thank reviewer #4 for his/her comments: 
The TTF1 negative group still need to a Napsin A for confirmation. Otherwise this study can be misleading.  -> We performed Napsin A on TTF-1 negative cases and introduced the following sentence in the results : " Napsin A was performed in TTF-1 negative patients, 3 of them were Napsin A positive."
We have modified this sentence in material and methods section : "Mucin stain was used to confirm the adenocarcinoma in case of negative TTF-1 and Napsin A (IP64, Leica) immunohistochemistry was perfomed in negative TTF-1."

This statement below need confirmation that the TTF-1 tumour is definitely lung adenocarcinoma. There is no basis so far to use molecular mutation as a modality to introduce a new group. 

"The different frequency of EGFR and ALK subtypes between TTF-1 positive and TTF-1 negative subtypes might be linked to the fact that TTF-1 negative tumors on biopsies represent a group enriched in rare adenocarcinomas subtypes. These subtypes are mostly represented by invasive mucinous adenocarcinoma representing roughly 10% of lung tumors and is often EGFR wild type and TTF-1 negative [2]."-> We have confirmed that these tumor were adenocarcinoma with mucin stain, TTF-1 or Napsin A

We hope you will find this manuscript suitable for publication.